# Treatment Pattern, Financial Burden, and Outcomes in Elderly Patients with Acute Myeloid Leukemia in Korea: A Nationwide Cohort Study

**DOI:** 10.3390/ijerph19042317

**Published:** 2022-02-17

**Authors:** Hyerim Ha, Yujin Jeong, Joo Han Lim, Young Ju Suh

**Affiliations:** 1Department of Internal Medicine, Inha University Hospital, Incheon 22332, Korea; ha.hyerim@snu.ac.kr; 2Department of Biostatistics, Korea University College of Medicine, Seoul 02841, Korea; wjddbwls1020@korea.ac.kr; 3Department of Biomedical Sciences, College of Medicine, Inha University, Incheon 22332, Korea

**Keywords:** acute myeloid leukemia, elderly, dose intensity, medical expenditure

## Abstract

Although approximately 50% of patients with acute myeloid leukemia (AML) are diagnosed over the age of 60 years, there is currently no established consensus on the treatment of elderly AML patients. Herein, we aimed to explore the incidence, medical expenditure, treatment, and outcomes of elderly AML patients in Korea by analyzing a nationwide cohort. We employed the Korean National Health Insurance Service-Senior cohort, which represents 10% of a random selection from a total of 5.5 million subjects aged 60 years or older. AML patients were identified according to the main diagnostic criteria of acute leukemia. Treatment for AML was divided into high- (high-dose cytarabine ± idarubicin) and low- (low-dose cytarabine or hypomethylating agents) intensity chemo-therapy and classified according to the chemotherapeutics protocol. We analyzed the survival outcomes and medical expenditures. Among 558,147 elderly patients, 471 were diagnosed with AML, and 195 (41.4%) were treated with chemotherapy. The median age was 65 years, and the median overall survival (OS) was 4.93 months (95% confidence interval, 4.47–5.43). Median OS was longer in patients undergoing chemotherapy than those in the best supportive care group (6.28 vs. 3.45 months, *p* < 0.001), and the difference was prominent in patients aged < 70 years. Twenty-eight (5.9%) patients received high-intensity chemotherapy, while 146 (31.0%) received low-intensity chemotherapy. The difference in median OS according to dose intensity was 4.6 months, which was longer in the high-intensity chemotherapy group (9.8 vs. 5.2 months in low-intensity group); however, the difference was not statistically significant. Patients who received high-intensity chemotherapy recorded longer hospital stays and incurred greater expenses on initial hospitalization. Elderly AML patients in Korea exhibited clinical benefits from chemotherapy. Although patients should be carefully selected for intensive treatment, chemotherapy, including low-intensity treatment, can be considered in elderly patients. Moreover, prospective studies on new agents or new treatment strategies are needed.

## 1. Introduction

Acute leukemia is one of the most commonly diagnosed cancers, accounting for about 3.5% of all cancer incidences and 4% of total cancer-derived mortalities in the United States, and is estimated to bear the 11th highest incidence among all cancers worldwide. The leukemia incidence has stayed relatively stable over the years, but regional variations have been reported in different geological areas owing to the differences in ethnicity, environmental factors, and lifestyles. Acute myeloid leukemia (AML) is the most common myeloid malignancy, accounting for 33.3% of all myeloid diseases in Korea [1]. The highest incidence of AML is observed in patients in their 70s, and approximately 50% of AML cases occur in patients aged 60 years or older. However, the survival outcomes of elderly patients with AML are inferior to those of young patients. The 5-year overall survival (OS) is less than 10% in patients over 70 years and 50% in patients under 50 years [2,3]. This age-dependent difference in prognosis is attributed to poor performance status, comorbidities, and increased early death caused by intensive chemotherapy [4]. Therefore, the first step in deciding which treatment to choose for elderly AML patients is to determine their fitness, and the single most important factor is the patient’s age at the time of diagnosis [5].

Since the European Organization for Research and Treatment of Cancer Leukemia Group trial demonstrated an increased survival rate on intensive induction chemotherapy compared to best supportive care in elderly AML patients, there have been studies to determine the standard regimen. Most studies enrolled patients over the age of 60 and attempted a treatment combination of cytarabine and anthracycline [6,7,8]. These studies showed that intensive chemotherapy could be an effective treatment option for selected elderly AML patients. Despite promising data on intensive chemotherapy, more than 50% of elderly AML patients show poor performance [9], leading to the need for an alternative treatment for unfit elderly patients. Low-dose cytarabine showed an improved complete response (CR) rate and increased OS in elderly patients [10]. Azacitidine and decitabine, which are hypomethylating agents (HMAs) for myelodysplastic syndrome, also demonstrated improved OS compared to conventional care, including low-dose cytarabine or induction chemotherapy [11]. In addition, new drugs for elderly AML patients were approved recently [12]. Targeted therapy with IDH1/2 (isocitrate dehydrogenase1/2) or FLT3 (fms-related tyrosine kinase3) presents a promising treatment option for elderly AML patients who have specific mutation targets [13]. Ivosidenib and enasidenib show effectiveness in IDH1- and IDH2-mutated AML, respectively [14,15]. Moreover, gilteritinib proves to be effective in FLT3-mutated AML [16]. However, the low incidence has rendered it challenging to conduct well-designed randomized controlled trials; therefore, a standard treatment for elderly patients with AML remains to be established. There is also no consensus on which patients should be treated with chemotherapy and what treatment should be administered. In this circumstance, evaluating what kind of treatment is being administered to elderly AML patients in real practice and the treatment outcome of each treatment will be necessary to develop appropriate treatment guidelines.

Real-world cost analysis for AML treatment is also vital for understanding the economic and physical resources necessary to improve management guidelines. For elderly AML patients, complication management from disease or medicine as well as supportive care are also essential, thus it is not easy to predict the economic burden from drug costs alone. Studies which assessed both direct and indirect costs of acute leukemia were relatively infrequent in the international literature. There is limited data regarding the economic burden associated with treatment of AML from time of diagnosis to death of a patient with AML.

This study aimed to explore the incidence, treatment pattern, and outcomes in elderly AML patients in Korea by analyzing a nationwide cohort of elderly Korean patients and determining which treatment should be considered. In addition, we evaluated the costs associated with AML in patients aged 60 years or older who were treated in routine practice in South Korea using an extensive nationwide cohort database.

## 2. Materials and Methods

### 2.1. Data Source and Definition

We used the Korean National Health Insurance Service-Senior (NHIS-Senior) cohort for this study analysis. This dataset was initially set up to provide high-quality longitudinal data about elderly patients that could be used to investigate their health outcomes and to predict risk factors in Korea [17]. This cohort represented 10% of a random selection from a total of 5.5 million subjects aged ≥ 60 years in the National Health Information Database (NHID) in December 2003 and followed up until December 2013 for all subjects, excluding those who expired. This database included personal data, disease diagnosis according to the International Classification of Disease tenth revision (ICD-10), death records, prescribed medical records, admission or emergency room (ER) visit history, and also included claimed total medical cost. The cohort protocol in this study was described previously in detail in another clinical study [17].

Our study defined patients with AML as having a main diagnosis corresponding with the ICD-10 code. (Appendix A) Patients who received treatment for AML were identified according to the code of chemotherapeutic agents administered in accordance with the diagnosis of leukemia. (Appendix A) The patients who diagnosed “newly” were selected from those whose AML was not registered in the claim for the previous 6 months. Treatment was classified into low- and high-intensity according to the initial treatment by reviewing the treatment history within 3 months of diagnosis. Low-intensity treatment was defined as chemotherapy with low-dose cytarabine or HMAs, such as azacitidine or decitabine. High-dose cytarabine ± idarubicin was classified as a high-intensity treatment. Promyelocytic leukemia, M3 according to the WHO classification of AML, was selected as a diagnosis code and claimed history of all-trans retinoic acid and excluded from the analysis of treatment outcomes as their treatment and prognosis were distinct from other subtypes of AML. Patients with a diagnosis of AML who received only supportive care including hydroxyurea or transfusion without active chemotherapy were considered as having received the best supportive care. OS was estimated from the date of diagnosis to the day of death or the last day on which insurance was claimed. Charlson comorbidity index (CCI) score, which is an index that reflects the severity and the patient’s health status, were evaluated based on the Korean standard classification of disease (KCD), including a total of 17 comorbidities based on ICD-10: myocardial infarction, congestive heart failure, peripheral vascular disease, cerebral vascular disease, dementia, chronic pulmonary disease, rheumatologic/connective tissue disease, peptic ulcer disease, mild liver disease, diabetes mellitus (DM) without chronic complications, paraplegia, renal disease, DM with chronic complications, any malignancy, moderate to severe liver disease, metastatic tumor, and acquired immune deficiency syndrome. A CCI score was calculated for each patient based on claims data, using evidence of the relevant KCD diagnoses during the 6-month period before the date of AML diagnosis. For each patient, CCI was measured for 0 to 17 according to comorbidities, and 0–1 points were classified as a low CCI and 2 or more were classified as a high CCI [18]. ER visit history of each patient, days of hospitalization, and medical expenditure were obtained by claims record of Health Insurance Review and Assessment, which merged on NHID during the study period.

### 2.2. Statistical Analysis

The chi-square (χ^2^) test was used to evaluate the independence of the categorical variables. An independent t-test was used to compare the differences in continuous variables. The Mann–Whitney U test was used to compare the median values of the biomarkers. The median OS was calculated using the Kaplan–Meier method, and comparisons of differences in survival rates between groups were assessed using log-rank tests. The Cox proportional hazard model was used to estimate hazard ratios (HRs) and 95% confidence intervals (CIs). Multiple Cox proportional hazard models were performed after adjusting for several factors known to be associated with the risk of death, such as age and CCI. The significance of all reported *p*-values was set at *p* < 0.05. All statistical analyses were performed using Statistical Analysis System version 9.4 (SAS Institute, Inc., Cary, NC, USA) and R Statistical Software version 4.0.2 (R Foundation for Statistical Computing, Vienna, Austria).

### 2.3. Ethics

The study protocol was reviewed and approved by the Institutional Review Board of Inha University Hospital (approval number: 2021-03-007-000). This study was conducted in accordance with the guidelines for biomedical research and the Declaration of Helsinki. The need for informed consent was waived by our institutional review board due to the study’s retrospective nature.

## 3. Results

### 3.1. Study Population

Among 558,147 subjects, 471 were diagnosed with AML during the follow-up period. A total of 195 patients treated with chemotherapy were administered cytotoxic and hypomethylating agents. Approximately 51% of treated patients were in their 60s, and the median age was 65 years. Approximately half of those who received chemotherapy had a low CCI, defined as a score of 0–1. Age and CCI were the factors that revealed statistically significant differences between the best supportive and chemotherapy groups. (Table 1).

### 3.2. Treatment Pattern and Outcomes

Of the patients diagnosed with AML, 41.4% were treated with chemotherapy, while the remaining 58.6% received only best supportive care. Twenty-eight (14.4%) patients received high-intensity treatment for AML, and 146 (74.9%) received low-intensity treatment. Approximately 94% (137/146) of patients with low-intensity treatment received low-dose cytarabine, while the other patients were treated with a hypomethylating agent. All-trans retinoic acid was administered to 21 (10.8%) patients with promyelocyte-type AML (M3) (Table 1).

The median OS of elderly AML patients was 4.93 months (95% CI, 4.47–5.43) (Figure 1a). When survival was analyzed by treatment, the patients who received chemotherapy, regardless of dose intensity, exhibited higher OS than those in the best supportive care group (6.28 months in chemotherapy vs. 3.45 months in best supportive care, *p* < 0.001). (Figure 1b) Among patients who had received chemotherapy, median OS was higher in the high-intensity treatment group than in the low-intensity group, but this was not statistically significant (*p* = 0.1008). (Table 2) Subgroup analysis between chemotherapy and supportive care group based on age and CCI revealed that chemotherapy-treated patients had a statistically significant OS benefit under 70 years of age (*p* = 0.0004, Figure 2a), and a higher OS in chemotherapy-treated patients in both the CCI < 2 (*p* = 0.035, Figure 3a) and CCI ≥ 2 (*p* < 0.001) subgroups. (Figure 3b) The patients aged ≥ 70 years showed an absolute OS gain of 1.7 months, although it was not statistically significant. (*p* = 0.138 shown in Figure 2b). As shown in Table 2, induction mortality, defined as death within eight weeks from diagnosis, was only observed in the low-intensity group (15.8%). The frequency of ER visits was 4.2 in the high-intensity treatment group and 3.7 in the low-intensity group. Among age, CCI, and chemotherapy, age represented the only factor that showed statistical significance for the hazards of death in both simple and multiple Cox proportional hazard models. After adjusting for CCI and chemotherapy, the hazards of death were 0.67 times lower in patients <70 years of age than in patients ≥ 70 years of age. (*p* = 0.0002, Table 3).

### 3.3. Medical Cost

Comparing the length of hospital stay, the patients who received a high-intensity AML chemotherapy regimen had a significantly longer hospital stay during the initial period of treatment than those who received low-intensity chemotherapy (*p* = 0.012). Comparing the costs of initial treatment, patients who received high-intensity chemotherapy incurred higher treatment expenses than patients who received low-intensity chemotherapy; however, this was not statistically significant. (Table 4 and Figure 4).

## 4. Discussion

In this population-based study, we analyzed the treatment patterns in patients aged 60 years and older who were diagnosed with AML between 2003–2013 in real practice in Korea. One of the major strengths of our analysis was the use of real-world survival data for all registered patients with AML in Korea for 10 years. Therefore, our results could represent all elderly patients with AML in Korea. There was a significant survival benefit in patients who received all types of anti-leukemic therapy, even among the low-intensity regimen and other therapy groups who had similar characteristics to the untreated group. In our study, the patients who received chemotherapy regardless of intensity showed prolonged survival of approximately 2.6 months compared to those who received the best supportive care only. This survival benefit was prominent in those under 70 years of age; however, approximately two months of survival gain was also confirmed in those aged 70 years or older. Additionally, regardless of CCI, patients who received chemotherapy lived longer than those who received supportive care. The findings from this study provide a rationale to strongly consider treatment with any kind of chemotherapeutic agent, rather than providing only the best supportive care in older patients who do not fulfill the criteria for receiving more intensive regimens. Prior clinical trials that included elderly AML patients support the results of our study [19,20].

Two extensive data analyses of elderly AML patients in the Surveillance, Epidemiology and End Results (SEER) Medicine database showed differences in treatment outcomes between chemotherapy and supportive care only. In a study by Meyers et al., they analyzed patients aged 65 years and older who were newly diagnosed with AML. This study included a total of 4058 patients between 1997 and 2007. Of these, only 43% of patients received chemotherapy, and 57% of patients were treated with supportive care only without chemotherapy. The median OS was 7 months in the chemotherapy group and 1.5 months in untreated patients [19]. The study also reported that elderly patients with more comorbidities were more likely not to receive chemotherapy and had shorter survival rates. Moreover, they reported AML as a disease that requires a substantial economic burden compared to only suboptimal improvement in treatment outcomes. A more recent study by Medeiros et al. analyzed 8336 patients aged more than 66 years diagnosed with AML between 2000 and 2009 from the SEER Medicare database. They showed that about 60% of patients still did not receive treatment after AML diagnosis, and the study confirmed that anti-leukemic agent treatment showed a significant survival benefit in elderly patients. Among AML patients older than 66 years, 40% received chemotherapy, and their 30-day mortality was 9%, which was lower than that of patients who did not receive chemotherapy [20]. In this study, as in our study, there was no information on performance status and baseline molecular or cytogenetic information for AML. In this study, intensive treatment revealed the greatest significant survival gain. Our study further solidifies these results and shows that intensive treatment should be a treatment of choice for elderly patients with few comorbidities. However, the two studies mentioned above included patients from 1997–2009, thus the patients in the studies were about 3–6 years earlier than our study. Therefore, a low-intensity regimen and supportive care such as antibiotics, antifungal agents, etc., is likely to be different from our study. Meanwhile, a population-based analysis of older adults with AML from the California Cancer Registry (CCR) between 2014 and 2017 were recently published [21]. Vanessa et al. analyzed 3068 patients with AML by treatment pattern (conventional vs. nonconventional), 80 years of age, comorbidities, ethnic backgrounds and public insurance. Among the patients, 64% of patients received conventional (42%) or nonconventional (22%) chemotherapy and their treatment outcomes proved superior to those without treatment. In our study, 58.6% of patients did not receive anti-leukemic chemotherapy treatment, and this portion appears similar to two previous studies with SEER data. However, in the analysis of the CCR data, the percentage of untreated patients reached as low as 22%, which may be attributed to the improvement of patient’s performance and the advance of supportive care with the passage of time. Among treated patients, the patients treated with conventional chemotherapy revealed increased survival rates than nonconventional chemotherapy treated patients. Due to the above studies which were conducted on United States patients, direct comparisons of studies require caution given the differences in regional, socio-economic and ethnic characteristics.

Studies on the treatment experience of elderly AML patients have shown that performance status bears a more significant effect on treatment outcomes than age itself [22,23]. Furthermore, with the advent of HMAs and other new chemotherapeutics, treatment strategies other than intensive chemotherapy have also been reported to improve survival [24,25,26]. Although these agents lead to a lower incidence of complete remission, they have a significant clinical benefit in decreasing disease progression and improving quality of life (QoL). Treatment with a low-intensity regimen was associated with improved survival and an approximately 20% reduction in the risk of death [27]. In our study, the patients who received HMAs comprised only 4.6%, while results of low dose cytarabine were 70.3%. This is due to the reimbursement of national insurance in Korea. Since HMAs in elderly AML patients were reimbursed in 2013, it is considered that a significant number of patients did not receive HMAs for economic reasons. Therefore, there may have been a change in the pattern of low-intensity treatment, thus additional studies are needed.

According to studies analyzing medical expenses in elderly AML patients, hospitalization-related expenses account for the most considerable portion of medical expenses, and lower treatment costs are used as the age increases [28,29]. In the cases of chemotherapy, survival rates were better even though hospitalization-related costs were increased and; hence, total medical cost increases. In a study comparing the cost-effectiveness of high-dose induction chemotherapy and decitabine, decitabine proved to be more cost-effective [30]. In our study, patients who received high-intensity chemotherapy recorded more extended hospital stays than patients who received low-intensity chemotherapy and, although not statistically significant with high range, incurred higher treatment-related expenses. It could provide another important clinical and financial benefit from low-intensity chemotherapy in elderly patients. Since there were limitations present in our cohort study, additional research on the financial difference according to treatment intensity is needed.

QoL represents another crucial outcome considered in treatment decisions among elderly patients with AML. Some studies have reported that QoL at the time of diagnosis was a result of the AML itself, and intensive chemotherapy could improve QoL, including early mortality, intensive care unit admission, fatigue, and physical function time [31,32]. Our study analyzed the frequency of ER visits and early mortality to evaluate QoL. The ER visits were higher in the patients receiving high-intensity chemotherapy, however, the difference was only 0.5, and there was no significant difference according to treatment intensity. Early mortality (<8 weeks) could not be compared between the low- and high-dose intensity groups due to the small number of patients receiving high-dose chemotherapy. Regarding the QoL in elderly AML patients, further studies based on individual patient data, such as patient-reported outcome measures, are needed.

Although this analysis of the Korean NHID is a nationwide claims database that could be used as a research tool to provide a basis for developing evidence-based policy interventions for cancer care, our study bears several limitations. A major limitation of nationwide claims data is the lack of genomic and clinical information. The NHID registry does not collect the baseline molecular and cytogenetic information of AML patients. We could not obtain information related to performance status, which could influence the decision of clinicians to select the optimal treatment strategy for elderly patients. Comorbidities in CCI were analyzed based on ICD-10 codes, which may be under or overestimated in some patients. Although our study was based on the Korean NHIS-Senior cohort, the number of analyzed patients was relatively small. However, regardless of these concerns, our study proves meaningful in that we have evaluated the role of chemotherapy in a general population group, including the long-term outcomes over a 10-year period.

## 5. Conclusions

In conclusion, elderly patients with AML experienced clinical benefits from chemotherapeutic agents in Korea. Our findings confirmed the benefit of treatment in contrast to palliative therapy in the elderly AML patient population and strongly indicated that chemotherapy, including a low-intensity regimen, should be considered in the majority of older patients. However, survival outcomes employing this approach to treatment remain dismal and warrant new agents or treatment strategies for older patients with AML. Our findings provide an important context for treatment decisions among older AML patients in Korea.

## Figures and Tables

**Figure 1 ijerph-19-02317-f001:**
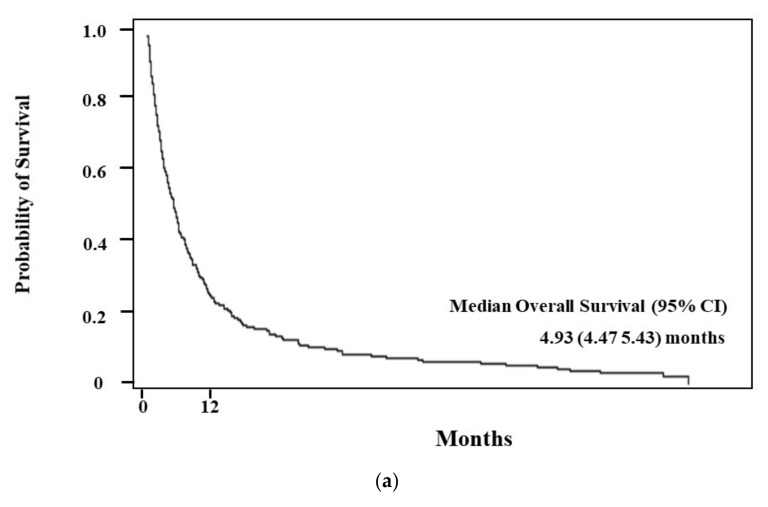
(**a**) Median overall survival of elderly acute myeloid leukemia patients was 4.93 months (95% CI: 4.47–5.43) months in Korea. (**b**) Median overall survival in chemotherapy group was 6.28 months (95% CI: 5.00–7.80), compared to 3.45 months (95% CI: 2.86–4.57) in best supportive care group.

**Figure 2 ijerph-19-02317-f002:**
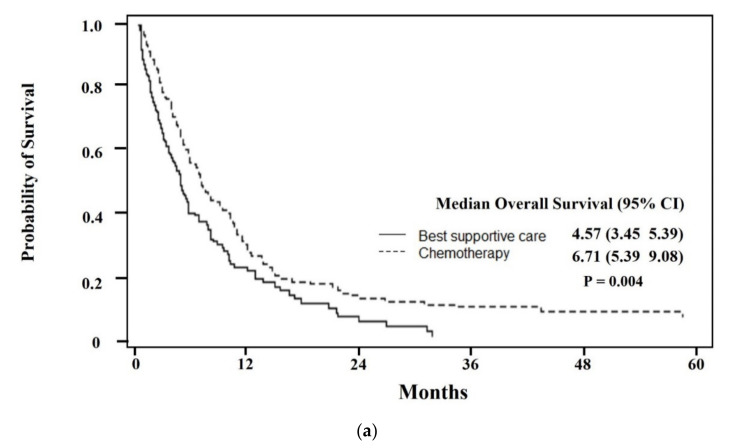
(**a**) Median overall survival for patients < 70 years of age treated with chemotherapy was 6.71 months (95% CI: 5.39–9.08), compared to 4.57 months (95% CI: 3.45–5.39) for patients < 70 years of age treated with best supportive care. (**b**) Median overall survival for patients ≥ 70 years of age treated with chemotherapy was 4.67 months (95% CI: 2.01–7.93), compared to 2.80 months (95% CI: 2.40–3.72) for patients ≥ 70 years of age treated with best supportive care.

**Figure 3 ijerph-19-02317-f003:**
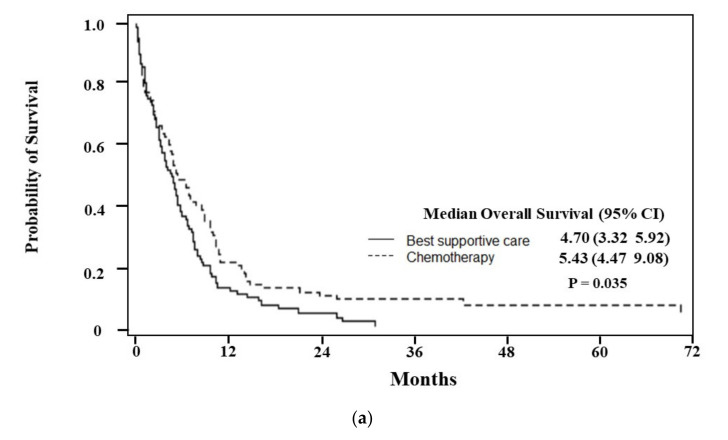
(**a**) Median overall survival for patients < 2 of Charlson Comorbidity index treated with chemotherapy was 5.43 months (95% CI: 4.47–9.08), compared to 4.70 months (95% CI: 3.32–5.92) for patients < 2 of Charlson Comorbidity index treated with best supportive care. (**b**) Median overall survival for patients ≥ 2 of Charlson Comorbidity index treated with chemotherapy was 6.51 months (95% CI: 5.13–8.75), compared to 2.83 months (95% CI: 2.47–3.98) for patients ≥ 2 of Charlson Comorbidity index treated with best supportive care.

**Figure 4 ijerph-19-02317-f004:**
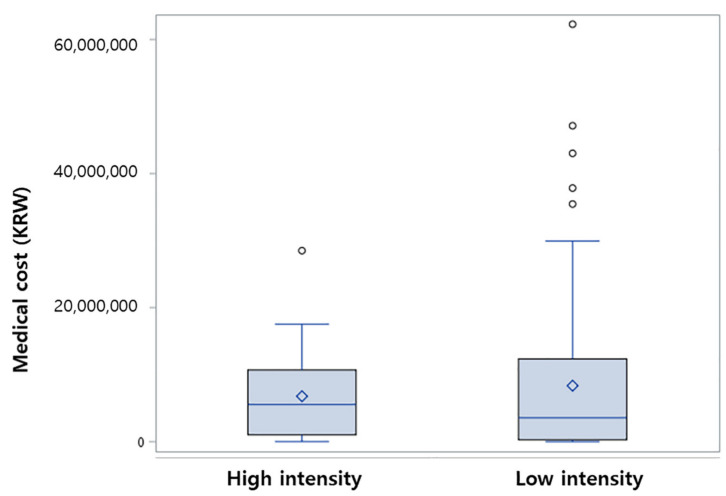
Medical cost by chemotherapy intensity. (blue line in the box: median of the cost; rhombus, mean of the cost; circle: outlier value like more than 1.5 times of upper quartile or less than 1.5 times of lower quartile).

**Table 1 ijerph-19-02317-t001:** Baseline characteristics.

Characteristics	Chemotherapy, N (%)	Best Supportive Care, N (%)	Total	*p* Value
All patients	195 (41.4)	276 (58.6)	471	
Treatment details				
High-intensity chemotherapy	28 (14.4)			
Low dose cytarabine	137 (70.3)			
Hypomethylating agents	9 (4.6)			
All-trans retinoic acid	21 (10.8)			
Hydroxyurea		77 (27.9)		
Best supportive care only		199 (72.1)		
Age				<0.0001
Median (IQR)	65 (62–69)	69 (65–73)		
60–69 years	148 (51.21)	141 (48.79)	289	
70–79 years	43 (26.06)	122 (73.94)	165	
≥80 years	4 (23.53)	13 (76.47)	17	
Gender				0.4247
Female	109 (43.08)	144 (56.92)	218	
Male	86 (39.45)	132 (60.55)	253	
CCI				0.0293
0–1	96 (47.06)	108 (52.94)	204	
≥2	99 (37.08)	168 (62.92)	267	

IQR, interquartile range; CCI, Charlson Comorbidity Index.

**Table 2 ijerph-19-02317-t002:** Clinical outcomes by dose intensity of treatment.

	High-Intensity Treatment	Low-Intensity Treatment	*p* Value
Total, N	28	146	
Frequency of ER visit (No. of ER visit/No. of patient)	4.214	3.712	
Induction mortality (<8 weeks), N (%)	0 (0)	23 (15.8)	
Overall survival, median (95% CI)	9.84 months (5.59–13.95)	5.23 months (4.47–6.97)	0.1008

ER, emergency room; CI, confidence interval.

**Table 3 ijerph-19-02317-t003:** Prognostic factors for the hazards of death.

	HR (95% CI)	*p* Value	Adjusted HR * (95% CI)	*p* Value
Age < 70	0.656 (0.537–0.801)	<0.0001	0.673 (0.547–0.828)	0.0002
CCI < 2	1.017 (0.835–1.238)	0.8690	1.010 (0.829–1.231)	0.9178
Chemotherapy	0.823 (0.674–1.005)	0.0555	0.911 (0.740–1.121)	0.3774

HR, hazards ration; CI, confidence interval; CCI, Charlson Comorbidity Index. * Adjusted HR, estimated hazard ratio of the covariate in the multiple Cox proportional hazard model.

**Table 4 ijerph-19-02317-t004:** Medical cost of initial hospitalization.

	High-Intensity Chemotherapy, Median (IQR)	Low-Intensity Chemotherapy, Median (IQR)	*p* Value
Duration (days)	37.0 (22–41)	22.5 (9–41)	0.0120
Total Cost (KRW, won)	5,540,645 (1,014,570–10,701,350)	3,566,285 (285,830–12,352,900)	0.8313

IQR, interquartile range; KRW, Korea Won.

## Data Availability

The data presented in this study are available on request from the corresponding author. The data are not publicly available because confirmation of the use of the original dataset is required by public institution.

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
