# Peer review of "Treatment Pattern, Financial Burden, and Outcomes in Elderly Patients with Acute Myeloid Leukemia in Korea: A Nationwide Cohort Study"

_ijerph, 2022, doi:10.3390/ijerph19042317_

Round 1

Reviewer 1 Report

In this study by Ha et al, the authors present an overview of the treatment options. clnical outcome and costs of hospitalization for elderly patients suffering from Acute Myeloid Leukemia (AML) in Korea.

The results should be better presented and described to the readership. 

  • Tables 1 and 2 should be merged to include the patients who had received jigh-intensit and low-intensity treatment. Abbreviations should be explained in the bottom of the Table.
  • Table 3 should be also reformated. The authors should add abbreviations, p values and add the word months to OS. 
  • In general, the quality of the figures should be improved.
  • Legends are absent from all figures
  • Most importantly though, it seems that there is inconsistency of what is described in the text and what is being presented in Figures 2 and 3. The Figures are descibed (in the text) to show a subgroup analysis of the chemotherapy treated group but they actually show comparisons between chemotherapy and best supportive care. 
  • Lines 63-65: "The recently developed..... who have specific mutation targets". Describe these mutations and give some examples from the litarature.
  • Lines 273-277: The results show high ranges in both high-intensity and low-intensity groups, so this judgement is somehow arbitrary.   

Minor corrections

Abstract: Line 26: add numbers instead of difference. Check the spelling in all abstract for example here-in, treat-ment etc 

Line 82: "in elderly AML patients". This is repetitive and should be removed. 

Line 154-158: Add percentages next to the numbers in each category

Line 162: Add the word "months" next to the numbers

Lines 195-198 and lines 204-209. They actually repeat the same thing.

Lines 250-253. Authors repeat almost the same thing in these two sentences. 

Author Response

Point 1. Tables 1 and 2 should be merged to include the patients who had received high-intensity and low-intensity treatment. Abbreviations should be explained in the bottom of the Table.

Response 1: Thank you for your comment. We merged Table 1 and 2, and add abbreviation in the bottom of the Table as your advice.

Point 2. Table 3 should be also reformatted. The authors should add abbreviations, p values and add the word months to OS. 

Response 2: Thank you for your comment. We reformatted Table 3 and add abbreviations, “months” and p value. 

Point 3. In general, the quality of the figures should be improved.

Response 3: Thank you for your comment. We revised the figure with improved quality.

Point 4. Legends are absent from all figures

Response 4: Thank you for your comment. We added figure legends.

Point 5. Most importantly though, it seems that there is inconsistency of what is described in the text and what is being presented in Figures 2 and 3. The Figures are described (in the text) to show a subgroup analysis of the chemotherapy treated group but they actually show comparisons between chemotherapy and best supportive care. 

Response 5: Thank you for your comment. Our results show that treatment outcomes difference between chemotherapy and best supportive care group. To clearly express our intentions, we have modified the sentence in results as below.

Line 177-184

Subgroup analysis between chemotherapy and supportive care group based on age and CCI revealed that chemotherapy-treated patients had a statistically significant OS benefit under 70 years of age (P=0.0004, Figure 2A), and a higher OS in chemotherapy-treated patients in both the CCI<2 (P=0.035, Figure 3A) and CCI≥2 (P<0.001) subgroups. (Figure 3B)

Point 6. Lines 63-65: "The recently developed .. who have specific mutation targets". Describe these mutations and give some examples from the litarature.

Response 6: Thank you for your comment. We added references and examples based on your advice

Line 64-69

In addition, new drugs for elderly AML patients were approved recently [12]. Targeted therapy with IDH1/2 (isocitrate dehydrogenase1/2) or FLT3 (fms-related tyrosine kinase3) is promising treatment option for elderly AML patients who have specific mutation targets [13]. Ivosidenib and Enasidenib show effectiveness in IDH1- and IDH2-mutated AML, respectively [14, 15]. Gilteritinib also effective in FLT3-mutated AML [16].

Point 7. Lines 273-277: The results show high ranges in both high-intensity and low-intensity groups, so this judgement is somehow arbitrary. 

Response 7: Thank you for your comment. We discussed that the difference in cost at first hospitalization or length of stay is meaningful, but it has a limitation of this study that it was not statistically significant with high range. Therefore, we added to suggest additional studies about financial difference between treatment.

Line 295-301

In our study, patients who received high-intensity chemotherapy had more extended hospital stays than patients who received low-intensity chemotherapy and, although not statistically significant with high range, incurred higher treatment-related expenses. It could be another important clinical and financial benefit from low-intensity chemotherapy in elderly patients. Since the limitation of cohort study, additional research on the financial difference according to treatment intensity is needed.

Minor corrections

Point 1 (Line 26): add numbers instead of difference. Check the spelling in all abstract for example here-in, treat-ment etc 

Response 1: Thank you for your comment. We add exact median OS in the sentence and cleaned the manuscript. 

Line 26-28

The difference in median OS according to dose intensity was 4.6 months, which was longer in the high-intensity chemotherapy group (9.8 vs. 5.2 months in low intensity group)

Point 2 (Line 82): "in elderly AML patients". This is repetitive and should be removed. 

Response 2: Thank you for your comment. We removed the phrase.

Point 3 (Line 154-158): Add percentages next to the numbers in each category

Response 3: Thank you for your comment. We add percentage next to the number.

Line 166-169

Twenty-eight (14.4%) patients received high-intensity treatment for AML, and 146 (74.9%) received low-intensity treatment. Approximately 94% (137/146) of patients with low-intensity treatment received low-dose cytarabine, while the rest were treated with a hypomethylating agent. All-trans retinoic acid was administered to 21 (10.8%)patients with promyelocyte-type AML (M3).

Point 4 (Line 162): Add the word "months" next to the numbers

Response 4: Thank you for your comment. We add the word “months” next to the number.

Line 174

6.28 months in chemotherapy vs. 3.45 months in best supportive care, P<0.001

Point 5 (Lines 195-198 and lines 204-209). They actually repeat the same thing.

Response 5: Thank you for your comment. We removed the sentence.

Point 6 (Lines 250-253). Authors repeat almost the same thing in these two sentences. 

Response 6: Thank you for your comment. We removed the sentence.

Reviewer 2 Report

I have reviewed the manuscript. This is a nicely performed study and a well written paper. I have a few comments/suggestions.

1) As far as I understand, the authors included patients diagnosed between 2003-2013. However, nowadays, there are many novel agents available in the management of AML (PMID: 31469910). I guess the authors should discuss this.

2) There is a recently published paper on the same issue (PMID: 34436782). I think the authors should cite this paper and discuss.

3) Although the authors stated that "Promyelocytic leukemia, M3 according to the WHO classification of AML, was selected as a diagnosis code and claimed history of all-trans retinoic acid and excluded from the analysis of treatment outcomes because of their treatment and prognosis were distinct from other subtypes of AML.", in part 3.2, they also stated that "All-trans retinoic acid was administered to 21 patients with promyelocyte-type AML (M3).". I guess since these patients were excluded from the outcome analysis, they should be omitted from all analysis.

4) Why do the authors think there were too little number of patients receiving HMAs? I guess this should be discussed.

5) Figure legends are missing. 

Author Response

Point 1. As far as I understand, the authors included patients diagnosed between 2003-2013. However, nowadays, there are many novel agents available in the management of AML (PMID: 31469910). I guess the authors should discuss this.

Response 1: Thank you for your kind comment. The development of new agents in elderly AML may have affected the results of this study, but since the details of each treatment was not covered in this study, the novel agent was dealt with in the introduction section. The article you recommended and its related contents have been added to the same part.

Line 65-69

In addition, new drugs for elderly AML patients were approved recently[12]. Targeted therapy with IDH1/2 (isocitrate dehydrogenase1/2) or FLT3 (fms-related tyrosine kinase3) is promising treatment option for elderly AML patients who have specific mutation targets[13]. Ivosidenib and Enasidenib show effectiveness in IDH1- and IDH2-mutated AML, respectively[14, 15]. Gilteritinib also effective in FLT3-mutated AML [16].

Point 2. There is a recently published paper on the same issue (PMID: 34436782). I think the authors should cite this paper and discuss.

Response 2: Thank you for your comment. We added the paper and additional discussion in discussion section.

Line 262-274

Meanwhile, a population-based analysis of older adult with AML from California Cancer Registry (CCR) between 2014 and 2017 were recently published [21]. Vanessa et al. analyzed 3,068 patients with AML by treatment pattern (conventional vs. nonconventional), 80 years of aged, comorbidities, ethnics and public insurance. Among the patients, 64% of patients received conventional (42%) or nonconventional (22%) chemotherapy and their treatment outcomes were superior to those without treatment. In our study, 58.6% of patients did not receive anti-leukemic chemotherapy treatment, and this portion looks similar previous two studies with SEER data. However, in the analysis of the CCR data, the percentage of untreated patients was as low as 22%, which may be attributed to the improvement of patient’s performance and the advance of supportive care with the passage of time. Because, among treated patients, the patients treated with conventional chemotherapy were increased than nonconventional chemotherapy treated patients

Point 3. Although the authors stated that "Promyelocytic leukemia, M3 according to the WHO classification of AML, was selected as a diagnosis code and claimed history of all-trans retinoic acid and excluded from the analysis of treatment outcomes because of their treatment and prognosis were distinct from other subtypes of AML.", in part 3.2, they also stated that "All-trans retinoic acid was administered to 21 patients with promyelocyte-type AML (M3).". I guess since these patients were excluded from the outcome analysis, they should be omitted from all analysis.

Response 3: Thank you for your comment. During the analysis of this study, our researchers discussed in-depth whether Promyelocytic leukemia (M3) should be included in the analysis. While other subtypes of AML showed a relatively similar disease course and treatment, Promyelocytic leukemia has unique disease character and different treatment. Therefore, M3 was not included in the analysis of treatment outcomes. However, since this study is also meaningful to look at the pattern of elderly AML patients in Korea, we decided to include M3 in the analysis of baseline characteristics. It would be helpful to understand elderly AML of Korea.

Point 4. Why do the authors think there were too little number of patients receiving HMAs? I guess this should be discussed.

Response 4: Thank you for your comment. It is important point. We think the cause is medical situation of Korea. HMAs were reimbursed in 2013 in Korea. Before then, patients should pay all medical cost while patients received low dose cytarabine pay only 5% of chemotherapeutics. We added this explanation in discussion section.

Line 284-289

In our study, the patients received HMAs were only 4.6 %, while low dose cytarabine is 70.3%. This is due to the reimbursement of national insurance in Korea. Because HMAs in elderly AML patients were reimbursed in 2013, it is thought that a significant number of patients did not receive HMAs for economic reasons. Therefore, there may have been a change in the pattern of low intensity treatment, so additional studies are needed.

Point 5. Figure legends are missing. 

Response 5: Thank you for your comment. We added figure legends.

Reviewer 3 Report

Major:

Could you define the best supportive care methods? Antibiotics, transfusions, hydroxyurea? I assume, some of these care options also require hospital visits. Is it possible to compare the medical costs of the best supportive care to high- and low-intensity chemotherapy? If yes, could you add it to Table 5 and Figure 4? If not, could you specify why?

Minor:

General:

Please remove dashes from some words, like pa-tients --> patients in line 14

All reference numbers and directions to figures and supplementary materials are placed after a punctuation mark. Please move all references and figure numbers before comma to clarify to which sentence it refers.

Materials and Methods:

I do not see supplements 1 and 2. Could you specify in the materials and method section if patients taken to analysis were newly diagnosed with AML? Please, specify also a time of treatment after diagnosis

Results

Please, specify figure numbers in lines:

160 --> Figure 1A

163 --> Figure 1B

To which figure refers the Figure 2 from line 167? Figure 2A?

Line 170 --> Figure 2B

I am not seeing any figure description. Overall title for the figure, subgroups chosen for generation of Kaplan-Meier plots should be placed below figures, and intensity of chemotherapy should be specified.

Author Response

Major:

Point 1. Could you define the best supportive care methods? Antibiotics, transfusions, hydroxyurea? I assume, some of these care options also require hospital visits. Is it possible to compare the medical costs of the best supportive care to high- and low-intensity chemotherapy? If yes, could you add it to Table 5 and Figure 4? If not, could you specify why?

Response 1: Thank you for your comment. We defined best supportive care as patients received no chemotherapy. For help clear understanding, this definition is added in methods section.

Line 114-116

Patients with a diagnosis of AML who received only supportive care including hydroxyurea or transfusion without active chemotherapy were considered to have received the best supportive care.

As your opinion, the best supportive care group also need hospital visit. So we compared frequency of ER visit only between low- and high-intensity treatment group because they may need hospital care or hospice care not ER.

Minor:

Point 1 (General)

Please remove dashes from some words, like pa-tients --> patients in line 14

All reference numbers and directions to figures and supplementary materials are placed after a punctuation mark. Please move all references and figure numbers before comma to clarify to which sentence it refers.

Response 1: Thank you for your comment. We cleaned the manuscript as your advice.

Point 2 (Materials and Methods)

I do not see supplements 1 and 2. Could you specify in the materials and method section if patients taken to analysis were newly diagnosed with AML? Please, specify also a time of treatment after diagnosis

Response 2: Thank you for your comment. The supplements 1 and 2 are below.

Supplement 1. Diagnostic code of acute myeloid leukemia according to the International Statistical Classification of Diseases tenth revision (ICD-10).

ICD-10 code

Diagnosis

C92

Myeloid leukemia

C93

Monocytic leukemia

C94

Other leukemias of specified cell type

C95

Leukemia of unspecified cell type

*  Excluding the diagnostic codes below

C92.1: chronic myeloid leukemia

C92.2: subacute myeloid leukemia

C92.3: myeloid sarcoma

C93.1: chronic monocytic leukemia

C94.4: acute panmyelosis

C94.5: acute myelofibrosis

C94.6: myelodysplasic and myeloproliferative disease, not classified

C95.1: chronic leukemia of unspecified cell type

Supplement 2. Code of chemotherapeutics agents according to the Anatomic Therapeutic Chemical Classification (ATC)

ATC code

Chemotherapeutics

139601BIJ, 139602BIJ, 139603BIJ, 139604BIJ, 139605BIJ, 139606BIJ, 139607BIJ, 139608BIJ, 139609BIJ, 139630BIJ, 139631BIJ, 139632BIJ, 139633BIJ, 139634BIJ, 139635BIJ, 139636BIJ, 139637BIJ, 139638BIJ, 139639BIJ, 139640BIJ

cytarabine

248001BIJ, 248002BIJ, 248003BIJ, 248030BIJ, 248031BIJ, 248032BIJ

vincristine sulfate

192101ATB, 192101BIJ, 192102BIJ, 192103BIJ, 92104BIJ,

192105BIJ, 192106BIJ, 192107ATB, 192107BIJ, 92108BIJ,

192109BIJ, 192110BIJ, 192111BIJ, 192112BIJ, 192132BIJ,

192134BIJ, 192136BIJ, 192138BIJ, 192139BIJ, 192140BIJ, 192141BIJ, 192142BIJ, 192143BIJ, 192144BIJ, 192145BIJ,

192146BIJ

methotrexate

190601ATB

mercaptopurine hydrate

181401BIJ, 181402BIJ, 181403BIJ

L-asparaginase

139001ATB, 139002BIJ, 139003BIJ, 139004BIJ, 139005BIJ

cyclophosphamide

196502BIJ, 196501BIJ, 196530BIJ, 196531BIJ

mitoxantrone

172001ACH, 172002ACH

hydroxyurea

237901ATB

thioguanine

243001ACS, 243002CCM, 243002COM, 243003CCM, 243003COM, 243004CCM, 243004CLQ, 243004COM, 243005CCM, 243009CLQ

tretinoin

495602BIJ, 495601BIJ

decitabine

484301BIJ, 484302BIJ

azacitidine

149430BIJ, 149431BIJ, 149432BIJ, 149433BIJ, 149434BIJ,

149435BIJ

doxorubicin hydrochloride

140601BIJ

daunorubicin hydrochloride

173001ACH, 173002ACH, 173002BIJ

idarubicin

The “newly diagnosed with AML” and “time of treatment after diagnosis” were specified in methods section as your advice.

Line 105-108

The patients who diagnosed “newly” were selected from those whose AML was not registered in the claim for the previous 6months. Treatment was classified into low- and high-intensity according to the initial treatment by reviewing the treatment history within 3 months of diagnosis.

Point 3 (Results)

Please, specify figure numbers in lines: 160 --> Figure 1A, 163 --> Figure 1B

To which figure refers the Figure 2 from line 167? Figure 2A? Line 170 --> Figure 2B

Response 3: Thank you for your comment. We added figure legends.

Point 4 (Results)

I am not seeing any figure description. Overall title for the figure, subgroups chosen for generation of Kaplan-Meier plots should be placed below figures, and intensity of chemotherapy should be specified.

Response 4: Thank you for your comment. We revised KM plots and chemotherapy included low- and high-intensity chemotherapy. Because the number of patients received high-intensity treatment was too small to analyze with low intensity chemotherapy group. We considered that to show the difference between chemotherapy group and best supportive group is more meaningful in this analysis.

Round 2

Reviewer 1 Report

The authors have replied to my comments.

I suggest that for figures 1-3, authors should have a united legend that would descibe what is seen in each part of the Figure (instead of adding a title underneath each part).

Authors should also check the manuscript for typos and speling errors 

Author Response

Thank you for your comment. We've inserted a united legend for Figures 1-3 with your advice. And we checked the manuscript for typos and spelling errors. 

This manuscript is a resubmission of an earlier submission. The following is a list of the peer review reports and author responses from that submission.